# Logical gates in ensembles of proteinoid microspheres

**Panagiotis Mougkogiannis**[ID]*, **Andrew Adamatzky**

Unconventional Computing Lab, University of the West of England, Bristol, United Kingdom

* Panagiotis.Mougkogiannis@uwe.ac.uk

## Abstract

Proteinoids are thermal proteins which swell into microspheres in aqueous solution. Ensembles of proteinoids produce electrical spiking activity similar to that of neurons. We introduce a novel method for implementing logical gates in the ensembles of proteinoid microspheres using chronoamperometry. Chronoamperometry is a technique that involves applying a voltage pulse to proteinoid microspheres and measuring their current response. We have observed that proteinoids exhibit distinct current patterns that align with various logical outputs. We identify four types of logical gates: AND, OR, XOR, and NAND. These gates are determined by the current response of proteinoid microspheres. Additionally, we demonstrate that proteinoid microspheres have the ability to modify their current response over time, which is influenced by their previous exposure to voltage. This indicates that they possess a capacity for learning and are capable of adapting to their environment. Our research showcases the ability of proteinoid microspheres to perform logical operations and computations through their inherent electrical properties.

**Data Availability Statement:** The data underlying the results presented in the study are available from https://zenodo.org/record/8286230.

**Funding:** The research was supported by EPSRC Grant EP/W010887/1 "Computing with

## Introduction

Unconventional computing methods, especially those based on organic substrates, are explored to address the limitations of current technology that heavily relies on silicon [1–3]. Great progress has been made in molecular and biomolecular logic systems [4–8]. These systems could serve as fundamental components for developing innovative types of computing in the future. The field of electrochemical logic gates has seen significant research activity [9–11]. Amatore and colleagues were among the early pioneers in this area, showcasing the capabilities of paired-band microelectrode assemblies. These assemblies successfully imitate the behaviour of neuronal synapses and are capable of performing Boolean logic operations. Amatore et al. demonstrated that artificial neurons utilising coupled double-band electrodes have the capability to operate as AND and OR logic gates [12, 13]. This is achieved by leveraging the distinctive diffusional cross-talk effects in close proximity to the electrodes. A detailed investigation was conducted on the time responses and theoretical features of these electrochemical logic gates. Expanding upon the encouraging progress made with paired microband electrodes as electrochemical logic gates, our current research focuses on the development of protenoid microsphere-based logic gates. Protenoid microspheres serve as a biomolecular platform that integrates the propagation, transmission, and detection of signals, similar to how natural neurons function.

proteinoids". The funders had no role in study design, data collection and analysis, decision to publish, or preparation of the manuscript.

**Competing interests:** The authors have declared that no competing interests exist.

Logic circuits can be developed by using synthetic molecules that can switch signals, as well as supramolecular systems, molecular machines [14], and nanoparticles that have been molecularly functionalized [15]. Researchers have developed biomolecular systems by using a variety of components, such as DNA, RNA [16], oligopeptides [17], proteins/enzymes [18], and biological cells [19]. These systems can operate within a living organism and are able to perform a range of logical operations/computational tasks, used as signal-responsive systems, logic functional devices, and binary YES/NO-operating biosensors. Many systems developed show some drawbacks related to repeatability and stability, time consuming implementation of I/O interface and not justifiably low speed of operations. This is why we propose to explore a unique class of organic devices—thermal proteins [20]—as a substrate and an architecture for future non-silicon massive parallel computers.

Thermal proteins (proteinoids) [20] are synthesized by thermal polycondensation of amino acids. This involves heating a mixture of amino acids to 160–200˚C under an inert atmosphere, triggering a polycondensation reaction between the amino acids. Rather than a typical polymerization which links monomer units together directly, this is a step-growth polymerization which also generates small molecule byproducts like water and ammonia. The high temperatures cause bifunctional amino acids like glutamic acid to cyclize, which facilitates their role as solvents and initiators for the polycondensation reaction. The end result is a complex mixture of polypeptides with a broad distribution of chain lengths [20, 21]. When a proteinoid is immersed in an aqueous solution at moderate temperatures (approximately 50˚C), it swells and forms structures known as microspheres [20]. These microspheres are hollow and typically contain an aqueous solution. The proteinoids are capable of folding into intricate shapes and interacting with different molecules, which makes them more versatile. Proteinoids are resistant and durable, they can withstand extreme temperatures and pH levels. Proteinoids can catalyse reactions and self-assemble into larger structures.

In [1] we proposed to prototype computing devices from the proteinoids. Such devices would be neuromorphic organic processors, which implement computing, via their electrical potential spiking activity. There is still limited knowledge about the capacity of proteinoid computers. In present paper we advance our knowledge of potential architectures of the proteinoid computers by using chronoamperometry technique and demonstrate feasibility of the approach with construction of several Boolean gates.

## Materials and methods

The present study used amino acids, namely L-Phenylalanine, L-Aspartic acid, L-Histidine, L-Glutamic acid, and L-Lysine, from Sigma Aldrich, which exhibited a purity level exceeding 98%. We synthesised proteinoid microspheres from amino acids by following the procedure depicted in Fig 1. The procedure consists of five steps:

- The amino acids weighing 5 g in total were heated to their boiling points and mixed together in equimolar amounts.

- The resulting mixture was then dissolved in water at a temperature of 80 degrees Celsius, while continuously mixing, to achieve a concentration of 10 mg/100 ml for each protenoid.

- The mixture was subjected to lyophilisation.

- After the lyophilisation process, the samples were collected.

- The collected samples were characterised using FT-IR and SEM techniques.

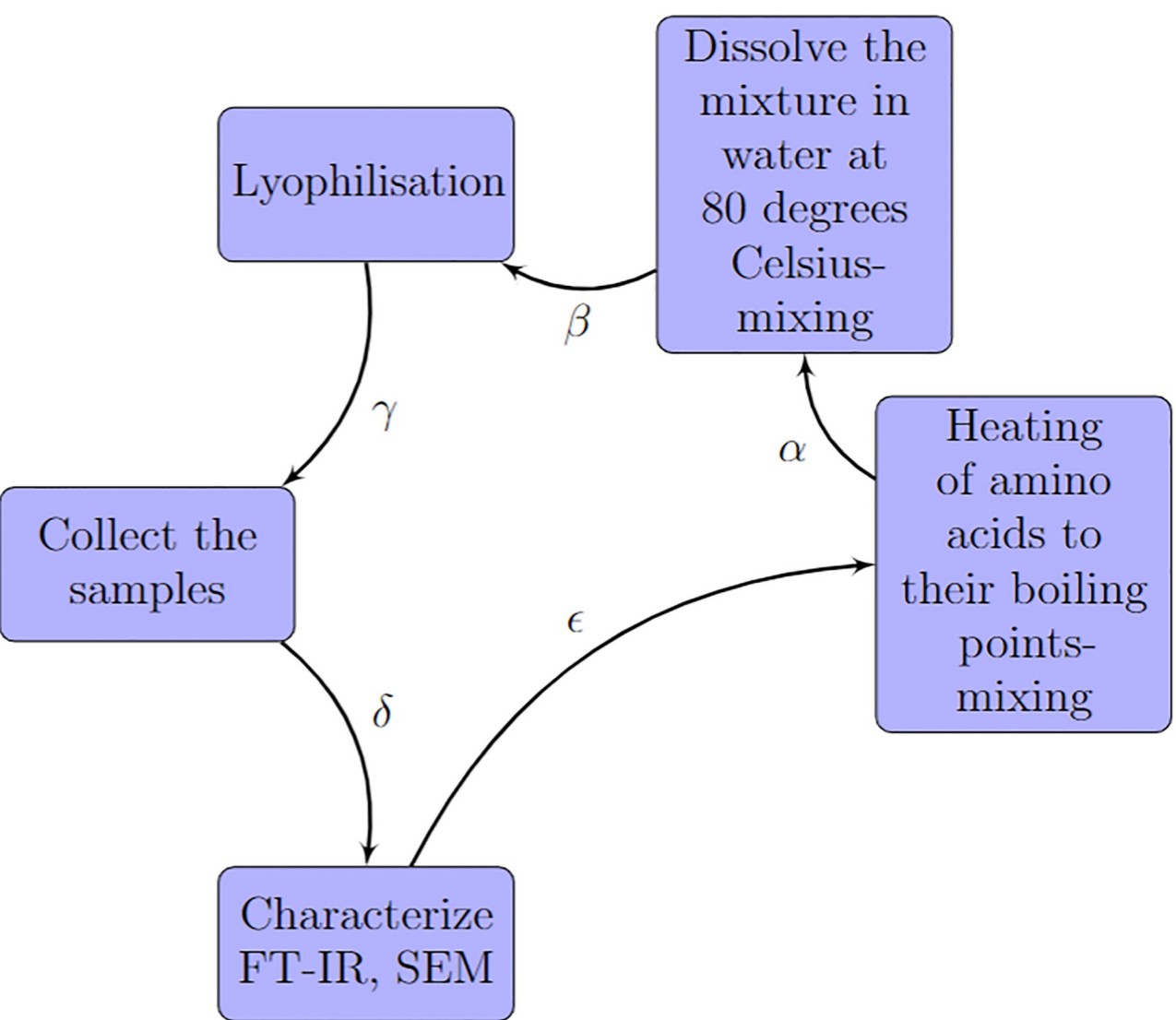

**Fig 1. Synthetic route for preparing protenoid microspheres.** The synthesis of proteinoid microspheres consists of five steps. A) Heat the amino acids until they reach their boiling points. B) Dissolve the mixture in water at a temperature of 80 degrees Celsius and mix thoroughly. C) Perform lyophilisation to remove any excess water. D) Collect the samples for further analysis. E) Characterise the samples using FT-IR and SEM techniques. The arrows in the diagram indicate the direction and order of the steps. Additionally, they are labelled with $\alpha$, $\beta$, $\gamma$, $\delta$, and $\epsilon$, which correspond to the reactions or transformations involved in each step. The figure was created using the tikzpicture environment in LaTeX.

The arrows in the diagram Fig 1 provide information about the direction and sequence of the steps. Additionally, they are labelled with symbols such as $\alpha$, $\beta$, $\gamma$, $\delta$, and $\epsilon$, which represent the reactions or transformations occurring in each step [22]. The proteinoids were characterised through Scanning Electron Microscopy (SEM) with the aid of FEI Quanta 650 instrumentation. The proteinoids were evaluated using Fourier-transform infrared spectroscopy (FT-IR) as well. The FT-IR spectra were acquired utilising a method that has been previously documented [22]. The Zimmer Peacock potentiostat Anapot EIS ZP1000080 was utilised for conducting chronoamperometry measurements.

We employed chronoamperometry as a technique to implement logical gates using proteinoid microspheres. We have prepared different types of proteinoids, labelled as A and B.

Proteinoids A and B are classified into groups based on their mean current values, which represent the average amount of electric charge passing through them. Based on this criterion, we can categorise the proteinoids into two distinct groups. Group A consists of proteinoids that exhibit positive mean current values, indicating a net flow of positive charge. The following combinations are: L-Glu:L-Asp:L-Phe, L-Lys:L-Phe:L-Glu, L-Glu:L-Phe:L-His, L-Glu: L-Phe: PLLA, L-Phe:L-Lys, and L-Glu:L-Asp:L-Pro. Group B consists of proteinoids that exhibit negative mean current values, indicating a net flow of negative charge. The following combinations are: L-Lys:L-Phe-L-His:PLLA, L-Glu:L-Arg, L-Asp, L-Phe, L-Glu:L-Phe, and L-Glu:L-Asp. A two-electrode setup was used for the electrochemical measurements, with platinum and irid-ium-coated stainless steel sub-dermal needle electrodes (Spes Medica S.r.l., Italy) serving as the working and counter/reference electrodes. The electrodes were positioned approximately 10 mm apart in a protenoid solution containing 10 mg/100 ml protenoid in water as the supporting electrolyte. The potentiostat applied a constant dc potential (Edc) and measured the current response at 0.1 s intervals for 25,000 s. For the results shown, Edc was 0.01 V, with no initial equilibration period. The proteinoids exhibited varying spiking frequency and amplitude when exposed to different applied potentials, specifically positive and negative current values. Communication with the protenoids was established using this two-electrode electrochemical cell connected to the potentiostat. We achieved different logic operations by modulating the spiking frequency and amplitude of the proteinoids through changes in the current. In this study, we present a potential mechanism for generating logical gates using proteinoid microspheres through the application of chronoamperometry.

The proposed mechanism is illustrated in Fig 2. This diagram in Fig 3 illustrates the potential of proteinoids as fundamental components of unconventional nanoscale computing.

## Results

The objective of the conducted experiments was to investigate the distinctive characteristics of proteinoids, including their capacity to generate microspheres, identify electrical signals, and serving as constituents of logic circuits. The ZP potentiostat was used to obtain chronoam-perometry measurements for all the proteinoids, as depicted in Fig 4. The device applies a steady direct potential (Edc) while the current is monitored at regular intervals (Fig 4).

The following examination will involve an exploration of the SEM features associated with proteinoid microspheres. The structure and orientation of proteinoids microspheres are depicted in Figs 5 and 6. The application of Scanning Electron Microscopy (SEM) imaging has uncovered the existence of proteinoids that comprise of nanospheres measuring 22.7 nm during embryonic phases (Fig 7). The nanospheres exhibit the ability to aggregate and subsequently amalgamate into larger clusters of microspheres measuring 5 microns in diameter. The negative image allowed for the identification of surface features and defects on the nanoparticles that were not initially observable in the original image. The images that were gamma-corrected revealed that increasing the gamma value resulted in improved brightness and contrast for the nanoparticles. However, it also introduced some noise and artefacts in the background.

The MATLAB algorithm utilises the `medfilt2` function to reduce noise in the image shown in Fig 5, from [24]. This function applies a median filter to the image, resulting in a cleaner and clearer output. When using this filter, each pixel value is substituted with the median value of the surrounding pixels. This technique is effective in reducing noise while also preserving edges. The image undergoes contrast enhancement through the use of the `imadjust` function. This function works by mapping the intensity values of the image to a new range, effectively adjusting its contrast. The programme utilises the edge function to detect edges in the image. Specifically, it uses the 'canny' method, which employs the Canny

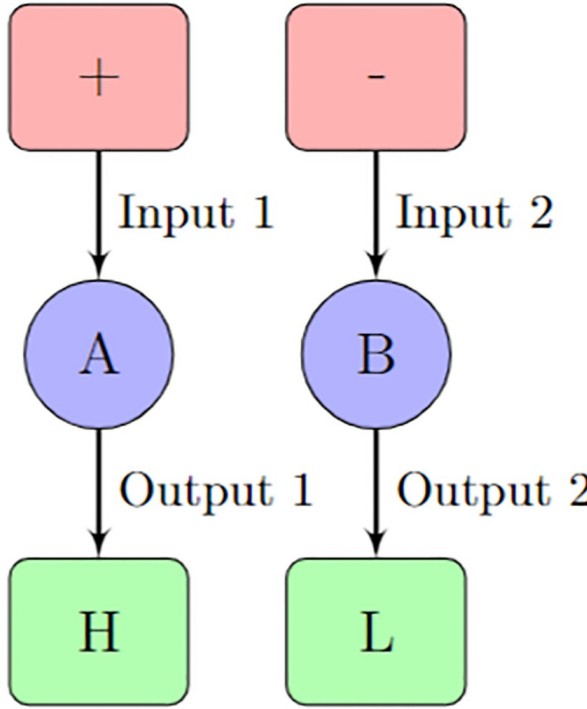

| A | Type A proteinoid |
| B | Type B proteinoid |
| + | Positive potential |
| - | Negative potential |
| H | High current response |
| L | Low current response |

**Fig 2. Controlling protenoid current signals for logic gating applications.** One potential approach for generating logical gates from proteinoid microspheres involves the utilisation of chronoamperometry. The figure displays two types of proteinoids, labelled as A and B, which exhibit distinct spiking frequencies and amplitudes in response to different potentials, specifically positive and negative potentials. We can achieve different logic operations by adjusting the spiking frequency and amplitude of the proteinoids through changes in the potential and electrolyte concentration. This allows us to obtain distinct current responses. This figure illustrates the implementation of an OR gate using type B proteinoids, a negative potential, and a high electrolyte concentration. The output signal will be determined by the current response of type B proteinoids.

algorithm to identify the edges present in the image. The algorithm employs a series of steps to achieve its goal, starting with smoothing and followed by gradient computation, non-maximum suppression, and hysteresis thresholding. After processing the input image, the output is a binary image that highlights the edges by assigning a value of 1 to those pixels and 0 to the rest. The code utilises the `imshowpair` function to showcase two images, the original and the edge image, side by side. This function creates a montage of the two images for easy comparison [25, 26].

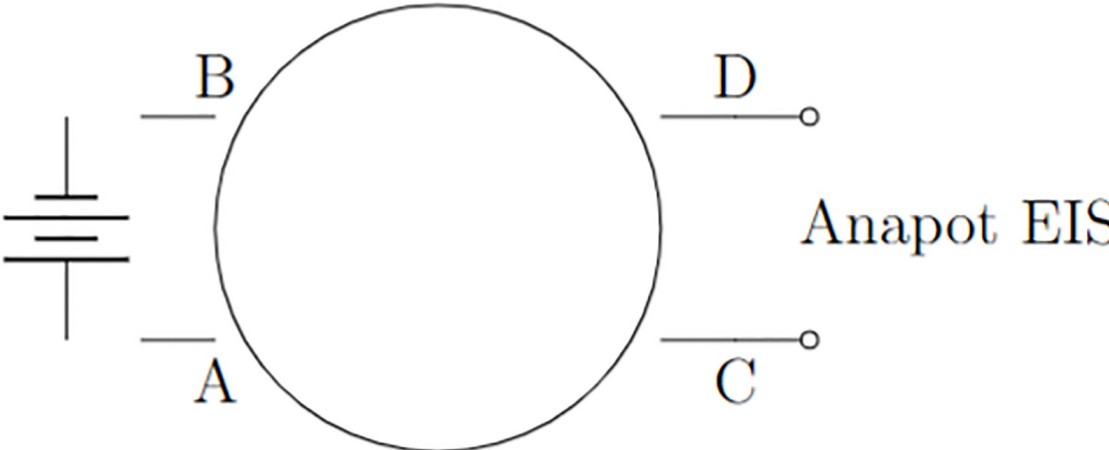

**Fig 3. Measuring protenoid microsphere current responses with applied potential.** A scheme of the experimental setup. An ensemble of proteinoid microspheres with four electrodes, depicted schematically. A voltage source is linked to electrodes A and B, while an Anapot EIS potentiostat is linked to electrodes C and D. The potentiostat detects and records electrical spikes induced in the microsphere by the voltage source.

Now, let us turn our attention to examining the data obtained from chronoamperometry. A boxplot is a graphical representation that displays the distribution of numerical data using five significant values: the minimum, maximum, median, lower quartile, and upper quartile. These values are used to provide a visual representation of the spread and central tendency of the data. The information displayed in the boxplot is as follows:

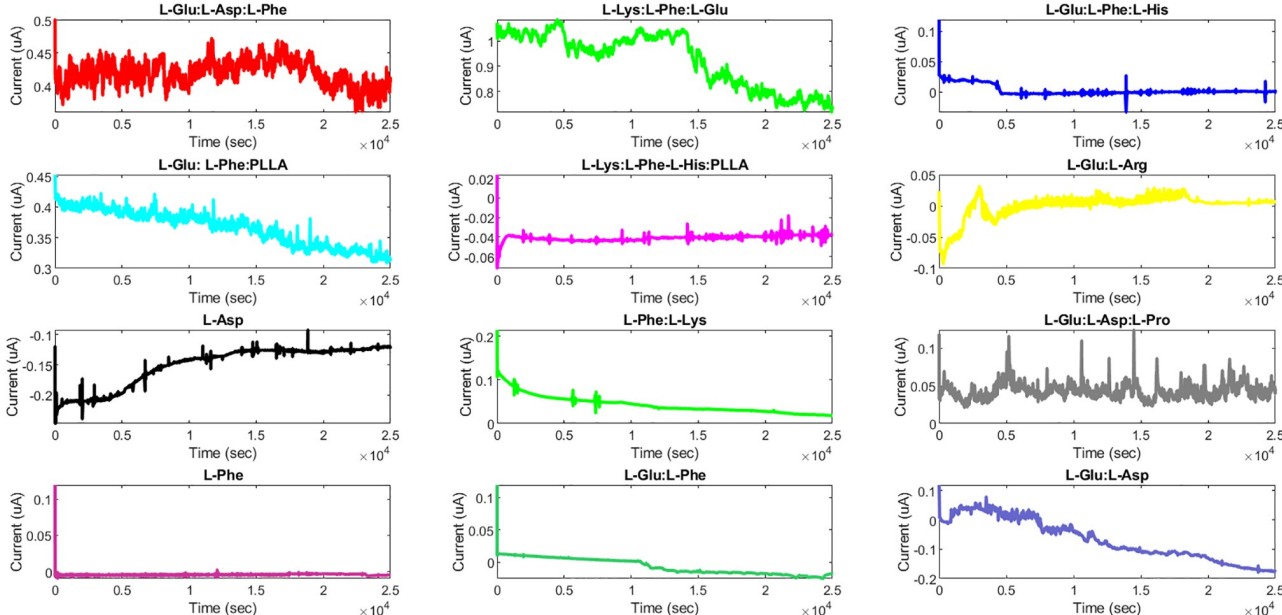

**Fig 4. Characterizing protenoid electroactivity using a ZP potentiostat.** The ZP potentiostat was used to obtain chronoamperometry measurements for all the proteinoids, as depicted in the figure. The device uses a steady direct current potential (E dc) while the current is monitored at regular intervals. To determine the concentration of a particular analyte in the sample, the obtained current is divided by a calibration factor [23].

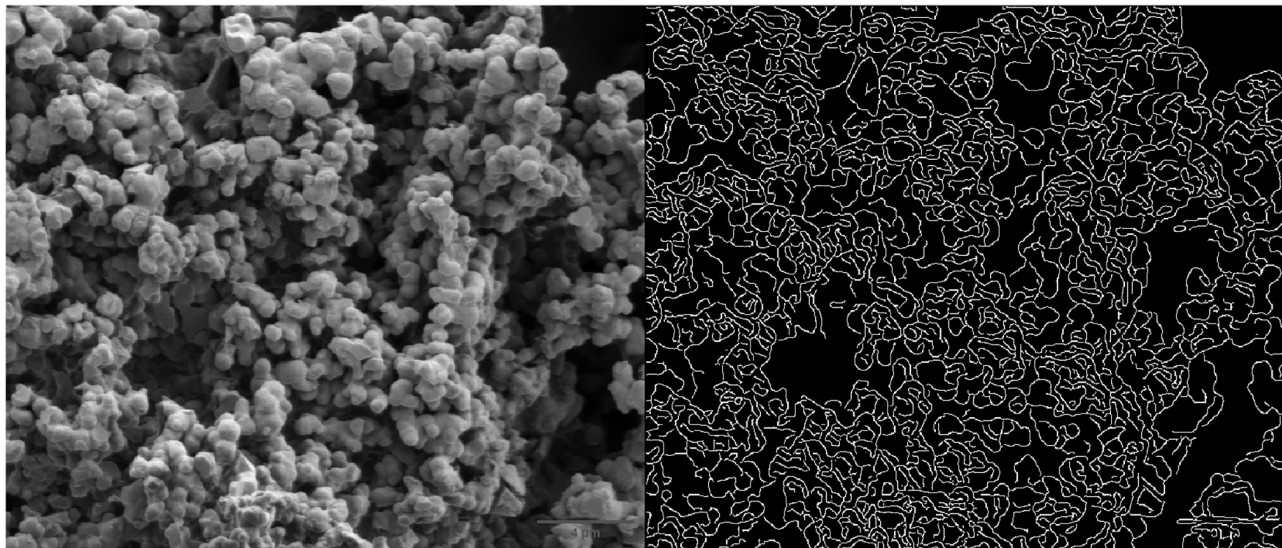

**Fig 5. Contrast, correlation, energy, and homogeneity metrics for protenoid edge detection.** After conducting an analysis on the edge detection of proteinoid nanospheres using Matlab, the results showed a contrast value of 0.1708, a correlation value of 0.9582, an energy value of 0.1410, and a homogeneity value of 0.9170.

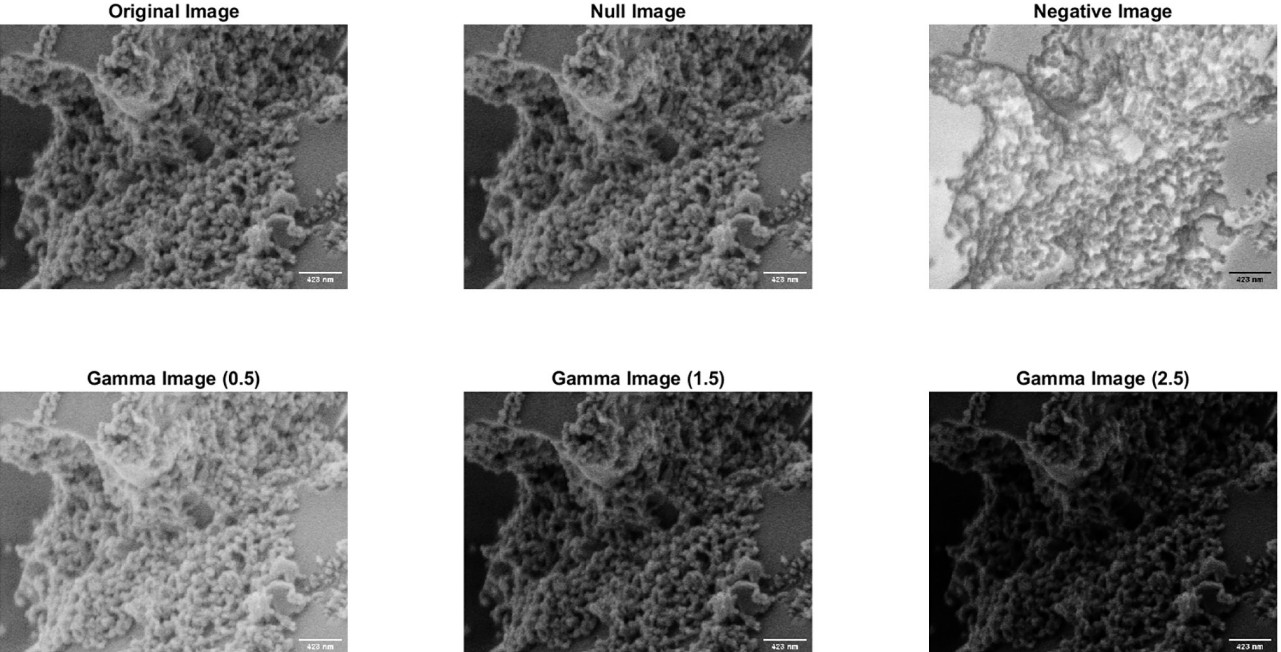

**Fig 6. Gamma correction for enhancing SEM imaging of protenoid nanoparticles.** The Impact of Gamma Correction on SEM Images of Proteinoid Nanoparticles. This is the original image of proteinoid nanoparticles captured using scanning electron microscopy (SEM) at 2 kV and 60,000x magnification. The negative image is created by inverting the grayscale values of the original image. The gamma-corrected images were obtained by applying different gamma values (0.5, 1.5, and 2.5) to the original image. Gamma correction is a technique used to adjust the brightness and contrast of an image by modifying its tonal range. The scale bar on the image corresponds to a length of 423 nanometers.

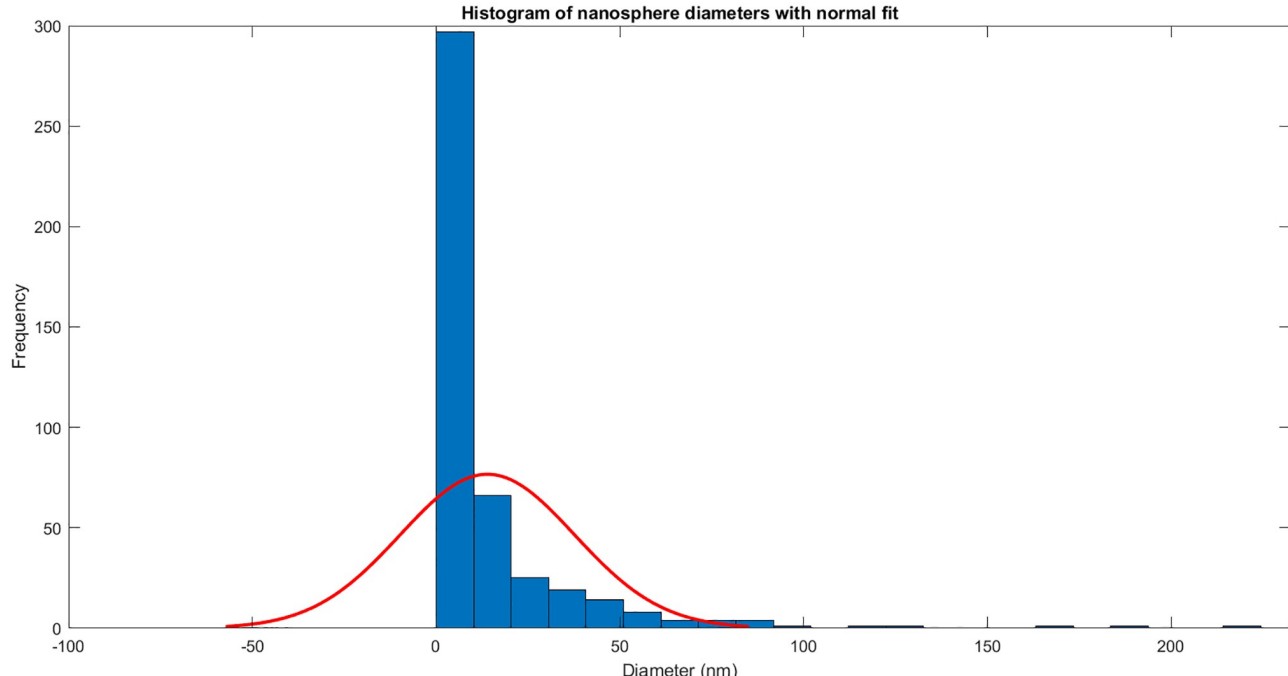

**Fig 7. Quantifying protenoid nanosphere size distribution.** The histogram depicts the frequency of proteinoid nanospheres with varying diameters, with $\mu$ = 13.9253 and $\sigma$ = 23.7247 representing a normal fit.

The box in a box plot is a visual representation of the middle 50% of the data. It spans from the lower quartile (Q1) to the upper quartile (Q3). The median (Q2) is represented by the line that runs through the centre of the box. This value corresponds to the middle number of the data set. The whiskers of a box plot represent the range of the data, from the minimum value to the maximum value, but they exclude any outliers. Individual points that fall outside the whiskers are represented as outliers on the plot. Outliers refer to data points that deviate significantly from the majority of the other data points, either by being exceptionally high or low in value. A boxplot is a useful tool for comparing various sets of data and determining their unique characteristics. It can assist in identifying key features of each group, including: the data's location by the median and the position of the box. One way to understand the spread of data is by looking at different measures such as the range, interquartile range (IQR), and the size of the box and whiskers. These measures can give us an idea of how much variability there is in the data. The skewness of the data refers to whether the box and whiskers plot is symmetrical or asymmetrical. Outliers are data points that fall beyond the whiskers, indicating their presence in the dataset. The current (in A) for each proteinoid was measured and compared using various statistical methods. Table 1 presents the interquartile range (IQR), skewness, kurtosis, and mean values of the current for each proteinoid. The proteinoid that exhibited the highest median current was L-Lys:L-Phe:L-Glu. This proteinoid also displayed the largest range and interquartile range (IQR), suggesting a significant variation in the current values. The proteinoid that had the lowest median current was L-Asp. It exhibited a negative skewness and kurtosis, indicating a left-skewed and platykurtic distribution. The proteinoid with the most pronounced skewness was L-Phe. It exhibited a significantly high positive skewness and kurtosis, suggesting a distribution that is skewed to the right and leptokurtic. The majority of proteinoids had a positive mean current, with the exception of L-Lys:L-Phe-L-His:PLLA, L-Glu:L-Arg, L-Asp, L-Phe, and L-Glu:L-Phe. The results indicate that various proteinoids

**Table 1. Summary statistics of current in A for each proteinoid.**

| Proteinoid | IQR | Skewness | Kurtosis | Mean |
|---|---|---|---|---|
| L-Glu:L-Asp:L-Phe | 2.6035e-08 | -0.17158 | 2.6377 | 4.17671e-07 |
| L-Lys:L-Phe:L-Glu | 2.1277e-07 | -0.42603 | 1.6076 | 9.24110e-07 |
| L-Glu:L-Phe:L-His | 2.7657e-09 | 1.728 | 5.271 | 2.79976e-09 |
| L-Glu: L-Phe:PLLA | 5.25e-08 | -0.25983 | 1.7499 | 3.64721e-07 |
| L-Lys:L-Phe-L-His:PLLA | 3.3855e-09 | -3.3672 | 34.9448 | -4.10408e-08 |
| L-Glu:L-Arg | 6.1989e-09 | -2.5045 | 9.6295 | -7.30825e-10 |
| L-Asp | 4.8208e-08 | -0.96935 | 2.3411 | -1.51220e-07 |
| L-Phe:L-Lys | 2.3317e-08 | 1.3707 | 5.107 | 4.30259e-08 |
| L-Glu:L-Asp:L-Pro | 1.2255e-08 | 1.0771 | 5.8677 | 4.21394e-08 |
| L-Phe | 1.0967e-09 | 29.4475 | 3858.3607 | -4.13615e-09 |
| L-Glu:L-Phe | 2.1982e-08 | 0.19427 | 1.6519 | -6.43642e-09 |
| L-Glu:L-Asp | 1.2573e-07 | 0.15665 | 1.6741 | -6.69806e-08 |

have varying effects on the current in the proteinoid microspheres. Additionally, certain proteinoids may function as logical gates, depending on the current's threshold.

We utilised MATLAB code to generate logical gates based on the chronoamperometry data of 12 distinct proteinoids that comprise proteinoid microspheres. We employed chronoamperometry to measure the current values of each proteinoid in response to external stimuli. Next, we utilised the average values of the current for each proteinoid as the threshold values to ascertain the presence or absence of a proteinoid in the sample. We performed four distinct logical operations (AND, OR, XOR, and NAND) on the binary matrix representing the presence or absence of each proteinoid. This allowed us to obtain the output for each logical gate. Fig 10 displays the time-dependent output of each logical gate. We noticed that the output patterns varied based on the logical operation and the threshold values.

Furthermore, we noticed that certain logical gates were more prone to being affected by noise and outliers in comparison to others. To assess the noise and outliers, we utilised the standard deviation and identified extreme values (those exceeding three standard deviations from the mean) of the output current for each logical gate. The OR gate exhibited a higher number of false positives compared to the AND gate, whereas the XOR gate demonstrated a higher number of false negatives in comparison to the NAND gate. The results indicate that it is possible to generate logical gates from proteinoid microspheres by analysing chronoamperometry data and utilising MATLAB code.

Additionally, we used MATLAB code to plot the current values and their distribution for each proteinoid. Fig 8 illustrates the distribution of current values for each proteinoid using boxplots. We noticed that some proteinoids displayed higher or lower levels of variation and outliers in their current values. Fig 9 illustrates the temporal progression of the proteinoid values. During our observation, we noticed variations in the current values of various proteinoids. Furthermore, we noticed that specific proteinoids exhibited comparable patterns of current fluctuations over time, whereas others demonstrated unique patterns. Furthermore, we discovered that certain proteinoids exhibited either skewed or symmetric distributions. Plots serve as valuable tools for comparing and contrasting the electrical properties of different proteinoids. They also help determine the optimal threshold values needed to create logical gates.

We used chronoamperometry data from 12 different proteinoids to create Boolean logic gates based on their presence or absence in a solution at a specific potential. We applied either a positive or a negative potential to the solution and measured the current flowing through each proteinoid at various time points. Next, we transformed the existing values into binary

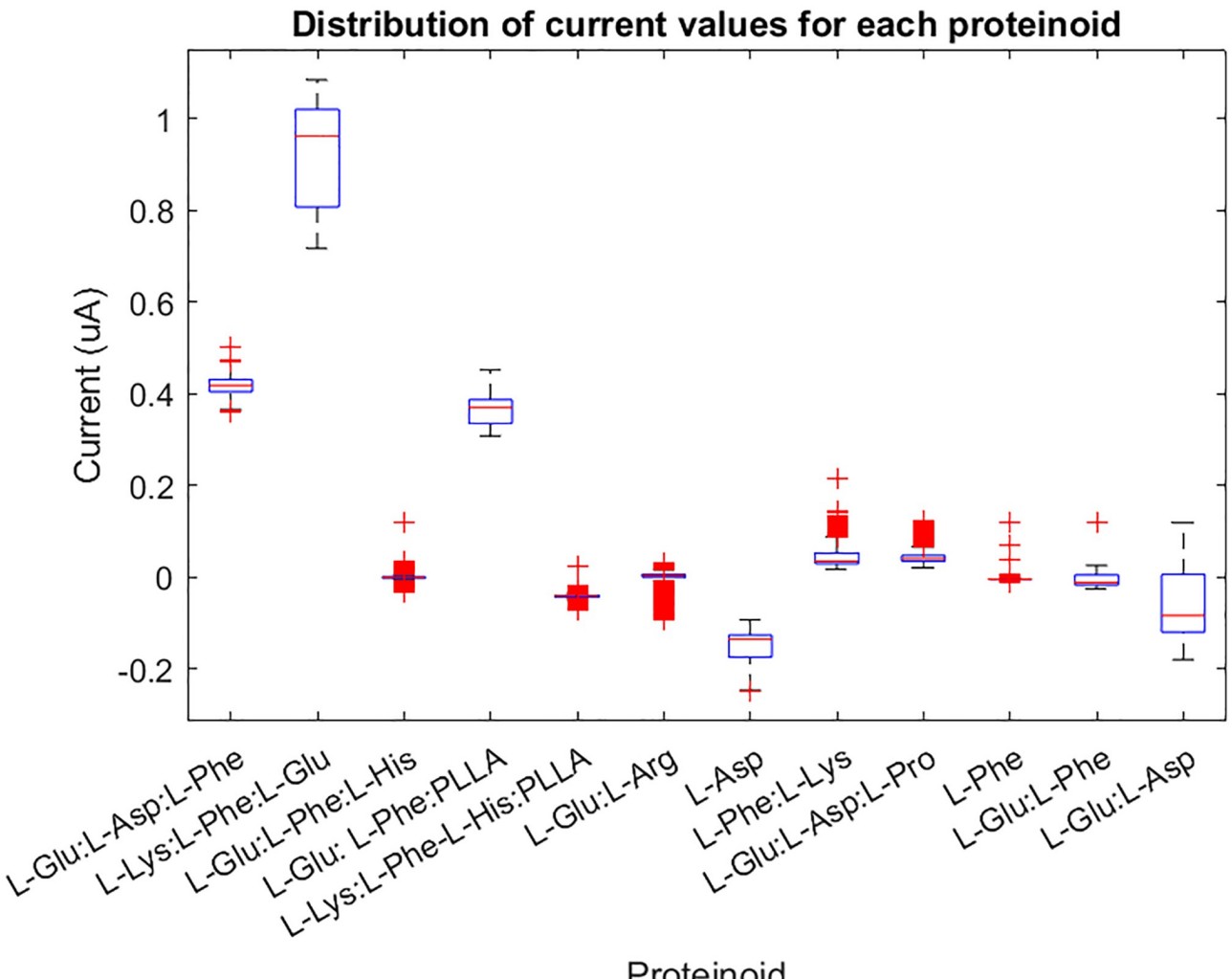

**Fig 8. Visualizing protenoid current distribution characteristics with boxplots.** The current values for each proteinoid are distributed. The boxplots display the median, interquartile range, and outliers of the current values for each proteinoids. The proteinoids names are displayed on the x-axis, while the current in amperes is shown on the y-axis. The figure depicts the skewness and kurtosis of the current values for each proteinoid.

values (0 or 1) using a threshold value. In this case, a value of 0 represents the absence of a proteinoid, while a value of 1 represents its presence. The threshold value for each proteinoid was determined by calculating the average current value across all time points. We applied the logical rules of AND, OR, XOR, and NAND to each pair of binary values in the proteinoid presence binary matrix, resulting in four logic operations being performed. For the AND operation, we assign a 1 to the output only if both binary values are 1; otherwise, we assign a 0. We performed this process for every pair of proteinoids and for each logic operation. Next, we plotted the output values for each logical gate over time and compared them to the expected output values determined by the logical operation.

The output of each logic gate is determined by measuring the electrical current at a specific time point after the inputs have been mixed. The current can be classified as either high or low, which corresponds to the binary logic values of 1 or 0. The output patterns of each logic gate match the expected truth tables, which are shown in Tables 2–5. The Fig 10 displays the changes in current over time for each logic gate.

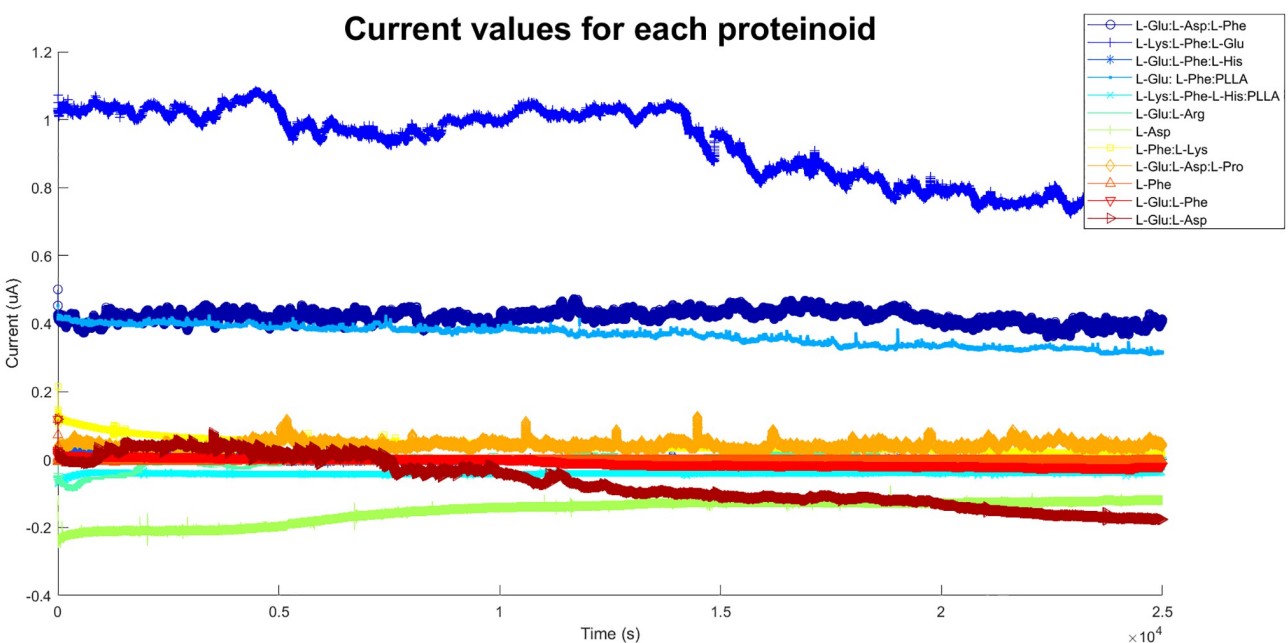

**Fig 9. Distinct current signatures of varied protenoid structures.** The figure displays the current values of 12 distinct proteinoids, which are responsible for the formation of proteinoid microspheres. The current values were measured using chronoamperometry in order to observe the response to external stimuli. Each protein is represented on the same graph using a distinct colour and marker. The x-axis represents time in seconds, while the y-axis represents current in amperes.

The truth table for the AND gate is displayed in Table 2. The output is 1 only when both inputs are 1; otherwise, it is 0. This indicates that the current is high only when both proteinoids L-Glu:L-Asp:L-Phe and L-Lys:L-Phe:L-Glu are present in the solution; otherwise, it remains low.

Table 3 displays the truth table for the OR gate. The output is 1 when either or both of the inputs are 1, and it is 0 otherwise. This indicates that the current is high when either or both of the proteinoids L-Glu:L-Phe:L-His and L-Glu:L-Phe:PLLA are present in the solution, and it is low otherwise.

Table 4 displays the truth table for the XOR gate. The output is 1 when only one of the inputs is 1, and 0 when either both or neither of them are 1. This indicates that the current is high when only one of the proteinoids, either L-Glu:L-Asp:L-Phe or L-Lys:L-Phe:L-Glu, is present in the solution. On the other hand, the current is low when both proteinoids are present or when none of them are present.

The truth table for the NAND gate is displayed in Table 5. The output is 0 when both inputs are 1, and 1 when either or neither of them are 1. This indicates that the current is low when both proteinoids L-Glu:L-Asp:L-Phe and L-Lys:L-Phe:L-Glu are present in the solution, and high when either one or none of them are present.

**Table 2. Truth table for the AND gate from proteinoid reactions.**

| Input A (L-Glu:L-Asp:L-Phe) | Input B (L-Lys:L-Phe:L-Glu) | Output (Current) |
|---|---|---|
| 0 | 0 | 0 |
| 0 | 1 | 0 |
| 1 | 0 | 0 |
| 1 | 1 | 1 |

**Table 3. Truth table for the OR gate from proteinoid reactions.**

| Input A (L-Glu:L-Phe:L-His) | Input B (L-Glu:L-Phe:PLLA) | Output (Current) |
|:---:|:---:|:---:|
| 0 | 0 | 0 |
| 0 | 1 | 1 |
| 1 | 0 | 1 |
| 1 | 1 | 1 |

**Table 4. Truth table for the XOR gate from proteinoid reactions.**

| Input A (L-Glu:L-Asp:L-Phe) | Input B (L-Lys:L-Phe:L-Glu) | Output (Current) |
|:---:|:---:|:---:|
| 0 | 0 | 0 |
| 0 | 1 | 1 |
| 1 | 0 | 1 |
| 1 | 1 | 0 |

**Table 5. Truth table for the NAND gate from proteinoid reactions.**

| Input A (L-Glu:L-Asp:L-Phe) | Input B (L-Lys:L-Phe:L-Glu) | Output (Current) |
|:---:|:---:|:---:|
| 0 | 0 | 1 |
| 0 | 1 | 1 |
| 1 | 0 | 1 |
| 1 | 1 | 0 |

## Discussion

In this paper, we utilised chronoamperometry data from 12 distinct proteinoids to construct Boolean logic gates through proteinoid interactions. We implemented four logic operations (AND, OR, XOR, and NAND) on the binary matrix representing the presence of proteinoids. We then measured the output current at a specific time point. We discovered that by using various combinations of proteinoids, we were able to generate distinct patterns of output current.

Our findings indicate that proteinoid reactions can be used to implement Boolean-based digital circuits and computing devices. In line with previous research, proteinoid microspheres exhibit electrical pulses resembling neuron action potentials [27] and can form networks with programmable growth [28]. These investigations suggest that proteinoid microspheres can serve as proto-neural networks and unconventional computing devices [29]. Our findings demonstrate that proteinoid reactions can also conduct logical operations on binary inputs and generate binary outputs. This implies that proteinoid reactions can be utilised to develop more complex and flexible computational systems based on Boolean logic.

Nevertheless, our study has limitations that must be addressed. First, we only used 12 varieties of proteinoids as inputs to our logic gates, which may not accurately represent the diversity and complexity of proteinoid reactions. Second, the output current was only measured at a specific time point, which may not have captured the dynamic changes and interactions of proteinoids over time. Thirdly, we only performed four elementary logical operations on our binary matrix, which may not be representative of the complete potential and functionality of proteinoid-based computing devices. Consequently, our findings should be interpreted with caution, and additional experiments are required to validate and expand them.

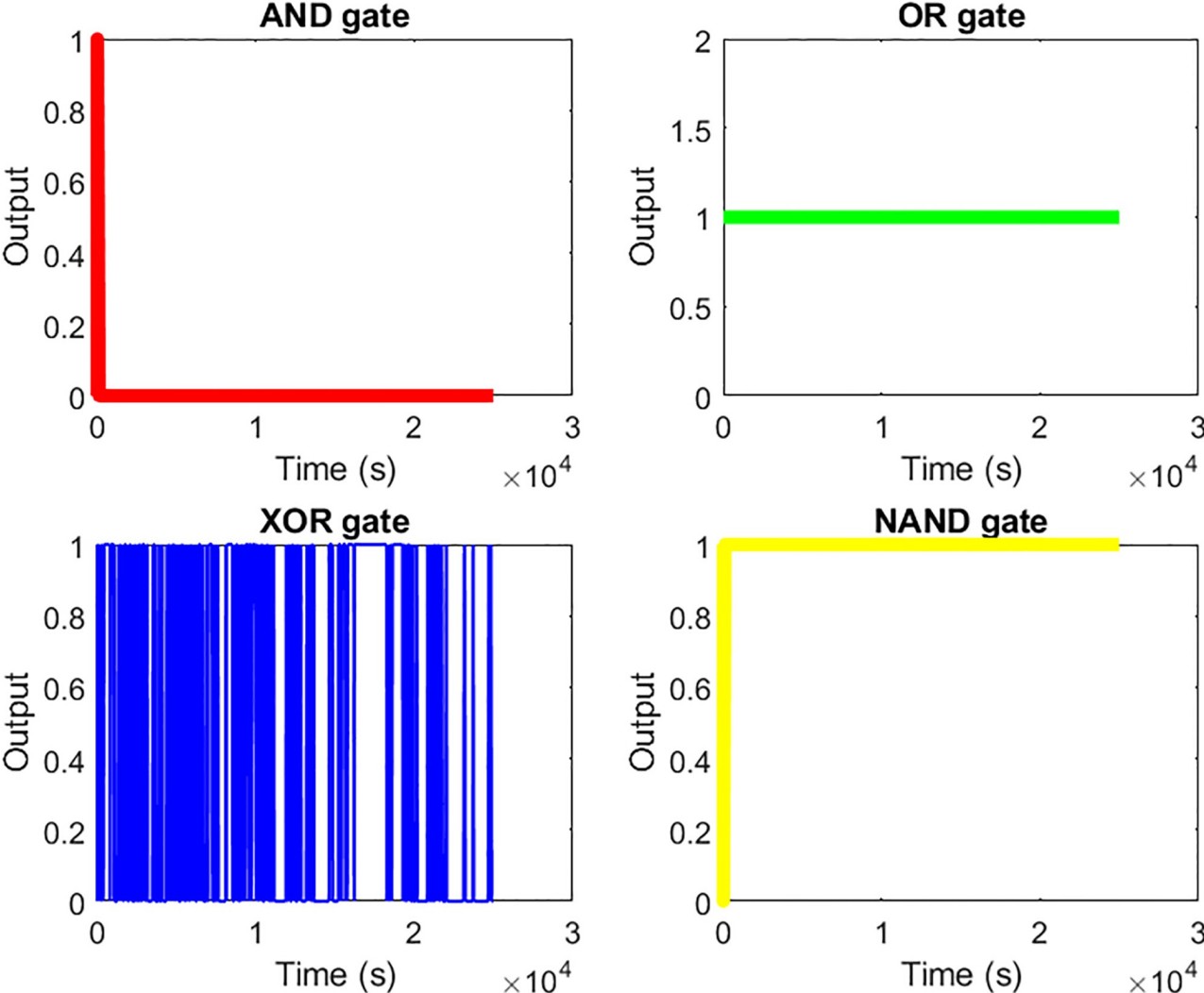

**Fig 10. Analysing chronoamperometry data to determine logical gates.** The figure displays the output of four distinct logical gates (AND, OR, XOR, and NAND) derived from the chronoamperometry data of 12 proteinoids. The output is a binary vector that indicates whether the logical condition is satisfied or not for each time point. The logical gates were constructed by using the average values of the current for each proteinoid as the threshold values. These threshold values were then used to determine whether each proteinoid was present or absent in the sample. Next, the binary matrix was subjected to logical operations such as AND, OR, XOR, and NAND in order to obtain the output of each gate. The output of each gate is displayed on a 2 by 2 grid of subplots, with each gate represented by a different colour. The AND gate is plotted in red, the OR gate in green, the XOR gate in blue, and the NAND gate in yellow. The x-axis represents time in seconds, while the y-axis represents the output, which can be either 0 or 1.

In order to enhance future research, we recommend further investigation into various proteinoids as potential inputs for logic gates. Additionally, exploring different reaction conditions and detection methods would be beneficial. We recommend measuring the output current at various time intervals to observe the temporal progression of proteinoid reactions. In addition, we recommend employing more complex logic operations and combining multiple logic gates to construct more complex proteinoid-based computing devices. These experiments have the potential to offer valuable insights into the properties and potential applications of proteinoid reactions in the field of bio-inspired computing.

The results indicate that proteinoid microspheres have the potential to serve as fundamental components for constructing artificial neural networks or bio-inspired computing devices.

**Table 6. Proteinoid microspheres, actin filaments, and polynucleotides in bioinspired computing.** The table shows biomolecule features and functions that can be used to create unique computing devices that mimic biological systems.

| Proteinoid microspheres | Actin filaments | Polynucleotides |
|---|---|---|
| • These structures are formed through self-assembly of thermal proteins, which are produced by heating amino acids [27]. | • Actin, a globular protein that is abundant in eukaryotic cells, forms polymers [37]. | • Polymers of nucleotides are the fundamental building blocks of DNA and RNA [38]. |
| • They are protoneurons because they can generate electrical impulses that resemble the action potentials of neurons [27]. | • They participate in a number of cellular processes, including muscle contraction, cell motility, cell division, and intracellular transport [37]. | • As ribozymes, they may store and transfer genetic information as well as perform catalytic tasks [38]. |
| • They are capable of forming networks with collective behaviour and emergent features such as synchronisation, memory, and learning [27]. | -They can interact with actin-binding proteins (ABPs), which control their dynamics and organisation[37]. | -They can self-assemble and hybridise to generate intricate shapes and patterns [38]. |
| • They can be used to construct bioinspired computing devices that replicate the operation of biological neural networks [27]. | • They can be employed as parts in bioinspired computing devices that take advantage of their mechanical and electrical features. | • They can be employed as components in bioinspired computing devices that take advantage of their data processing and catalytic capabilities [38]. |

One potential approach is to draw a comparison between our proteinoid microspheres and actin filaments. Actin filaments are natural protein polymers that have the ability to produce electrical signals when exposed to osmotic stress [30, 31]. As mentioned in the introduction, actin filaments play a crucial role in various cellular functions, including cell division, movement, and communication. Additionally, they play a role in the development of synapses and neural networks within the brain. Actin filaments can be seen as an illustration of how biological systems utilise the physical and chemical properties of proteins to encode and transmit information.

Another potential viewpoint is to establish a connection between our proteinoid microspheres and polynucleotides. Polynucleotides, whether synthetic or natural nucleic acid polymers, have the ability to generate electrical pulses under specific circumstances as well. In his influential paper, Manning proposed that polynucleotides have the potential to function as polyelectrolytes. These polynucleotides can generate electrical activity in response to variations in the salt concentration or pH level of the solution. Additionally, he proposed that these electrical pulses might serve as a rudimentary means of transmitting information in prebiotic systems. Polynucleotides serve as an example of how molecular systems utilise electrostatic interactions to store and manipulate information. The properties and functions of proteinoid microspheres, actin filaments, and polynucleotides, as presented in Table 6, can be utilised to develop innovative computing devices that imitate biological systems. One possible experimental approach is to use an electrochemical cell with three electrodes to conduct chronoamperometry measurements of protenoid solutions. The proposed setup is illustrated in Fig 11. The use of a potentiostat in this proposed setup would allow for the characterization of electrical spiking activity as predicted by Manning's theory.

By comparing and contrasting proteinoid microspheres, actin filaments, and polynucleotides, we can gain insights into the similarities and differences between artificial and natural information processing systems. Additionally, we can investigate the potential mechanisms and pathways that have contributed to the development and advancement of information processing in living organisms. We are hopeful that our work will serve as a catalyst for additional research and foster meaningful discussions on this particular topic [32–36].

Although our analysis primarily focuses on using single current measurements for implementing logic gates, we acknowledge that the protenoid signals display time-dependent variability. Distinct time scales are observed for the spiking behaviour and noise characteristics of different proteinoid types. The variability in the signals over time has the potential to affect the

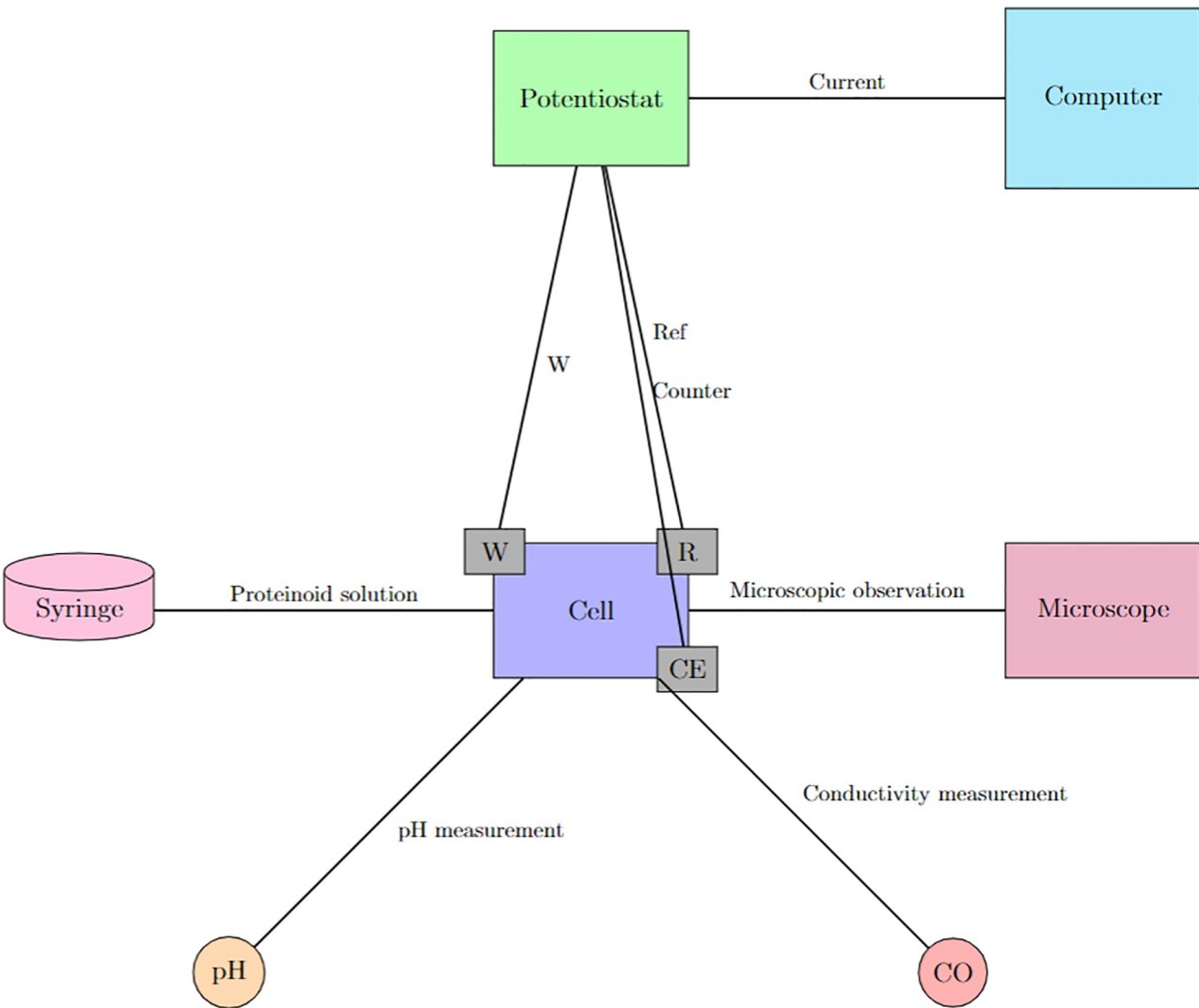

**Fig 11. Proposed apparatus to evaluate manning's hypothesis on protenoid signaling.** One potential experimental setup for testing Manning's 1978 theory with proteinoid solutions involves the use of chronoamperometry. The diagram illustrates an electrochemical cell comprising three electrodes: a working electrode (W), a reference electrode (R), and a counter electrode (CE). The cell contains an electrolyte solution and is connected to a potentiostat. The potentiostat applies a constant potential to the electrodes and measures the current response of the proteinoids present in the solution. The potentiostat is connected to a computer, which is responsible for recording and analysing the data. The syringe pump is used to inject a small quantity of proteinoid solution into the cell in close proximity to the working electrode. The pH metre and conductivity metre are used to measure the pH and conductivity of the proteinoid solution. The purpose of a microscope is to observe the morphology and size of proteinoid microspheres.

reliability and precision of logic operations, especially when using proteinoids with significantly different dynamics. Logic gating that relies on transient spikes may be impacted if the timing of the spikes is inconsistent. In order to obtain consistent logic inputs from protenoid currents that have a high degree of variability, it may be necessary to employ suitable signal processing or filtering techniques. Further investigation is needed to understand the impact of signal noise and fluctuations on logic accuracy. The present study showcases the use of simple DC signal levels to perform proof-of-concept logic operations. This serves as a starting point for further investigation into more intricate signal processing methods that can help reduce time-dependent variations.

## Conclusion

To sum up, the utilisation of proteinoid microspheres presents a distinctive avenue for unconventional computing techniques, facilitating the execution of intricate logical operations and computational tasks. The application of chronoamperometry has enabled the construction of logical gates using proteinoid microspheres, thereby presenting an exciting possibility for the advancement of computing technologies.

## Supporting information

**S1 File. Excel file containing the raw data of chronoamperometric measurements of L-Asp proteinoid in aqueous solutions.** The current (in $\mu$A) is reported as a function of time (in s) under an applied potential.
(XLSX)

**S2 File. Excel file containing the raw data of chronoamperometric measurements of L-Glu: L-Arg proteinoid in aqueous solutions.**
(XLSX)

**S3 File. Excel file containing the raw data of chronoamperometric measurements of L-Glu: L-Asp proteinoid in aqueous solutions.**
(XLSX)

**S4 File. Excel file containing the raw data of chronoamperometric measurements of L-Glu: L-Asp:L-Phe proteinoid in aqueous solutions.**
(XLSX)

**S5 File. Excel file containing the raw data of chronoamperometric measurements of L-Glu: L-Asp:L-Pro proteinoid in aqueous solutions.**
(XLSX)

**S6 File. Excel file containing the raw data of chronoamperometric measurements of L-Glu: L-Phe proteinoid in aqueous solutions.**
(XLSX)

**S7 File. Excel file containing the raw data of chronoamperometric measurements of L-Glu: L-Phe:L-His proteinoid in aqueous solutions.**
(XLSX)

**S8 File. Excel file containing the raw data of chronoamperometric measurements of L-Glu: L-Phe:PLLA proteinoid in aqueous solutions.**
(XLSX)

**S9 File. Excel file containing the raw data of chronoamperometric measurements of L-Lys: L-Phe:L-Glu proteinoid in aqueous solutions.**
(XLSX)

**S10 File. Excel file containing the raw data of chronoamperometric measurements of L-Lys:L-Phe:L-His:PLLA proteinoid in aqueous solutions.**
(XLSX)

**S11 File. Excel file containing the raw data of chronoamperometric measurements of L-Phe proteinoid in aqueous solutions.**
(XLSX)

**S12 File. Excel file containing the raw data of chronoamperometric measurements of L-Phe:L-Lys proteinoid in aqueous solutions.**
(XLSX)

## Acknowledgments

Authors are grateful to David Paton for helping with SEM imaging and to Neil Phillips for helping with instruments.

## Author Contributions

**Conceptualization:** Panagiotis Mougkogiannis, Andrew Adamatzky.

**Data curation:** Panagiotis Mougkogiannis.

**Funding acquisition:** Andrew Adamatzky.

**Software:** Panagiotis Mougkogiannis.

**Supervision:** Andrew Adamatzky.

**Writing – original draft:** Panagiotis Mougkogiannis.

**Writing – review & editing:** Andrew Adamatzky.

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
