## [Decision Letter · Decision Letter 0]

24 Aug 2023

PONE-D-23-22314Logical gates in ensembles of proteinoid microspheresPLOS ONE

Dear Dr. Mougkogiannis,

Thank you for submitting your manuscript to PLOS ONE. After careful consideration, we feel that it has merit but does not fully meet PLOS ONE’s publication criteria as it currently stands. Therefore, we invite you to submit a revised version of the manuscript that addresses the points raised during the review process. We would be happy to consider a revised version of the manuscript provided that you satisfy the remaining concerns of the reviewers. We understand that some revisions take time, but I should mention that we take into account the published literature available on the day we make our final decision.

We look forward to receiving your revised manuscript.

Kind regards,

Jianhui Liu

Academic Editor

PLOS ONE

Journal Requirements:

   "The research was supported by EPSRC Grant EP/W010887/1 “Computing with proteinoids”. Authors are grateful to David Paton for helping with SEM

imaging and to Neil Phillips for helping with instruments."

   "The research was supported by EPSRC Grant EP/W010887/1 “Computingwith proteinoids”.

4. Please expand the acronym “EPSRC” (as indicated in your financial disclosure) so that it states the name of your funders in full.

6. Please ensure that you refer to Figure 4 and 11 in your text as, if accepted, production will need this reference to link the reader to the figure.

Reviewers' comments:

Reviewer's Responses to Questions

**Comments to the Author**

1. Is the manuscript technically sound, and do the data support the conclusions?

Reviewer #1: Yes

Reviewer #2: Yes

2. Has the statistical analysis been performed appropriately and rigorously? 

Reviewer #1: Yes

Reviewer #2: Yes

3. Have the authors made all data underlying the findings in their manuscript fully available?

Reviewer #1: Yes

Reviewer #2: Yes

4. Is the manuscript presented in an intelligible fashion and written in standard English?

Reviewer #1: Yes

Reviewer #2: Yes

5. Review Comments to the Author

Reviewer #1: The paper is indeed excellent !!! It opens completely new aspects of the biomolecular computing. In addition to the fundamental scientific value, the paper offers novel options for practical applications, particularly in the field of biomedical use. It is strongly recommended for urgent publication after some minor (mostly formatting) editing.

The following changes are recommended for improving the paper, which is scientifically excellent:

1. All figures must be improved for better visibility (increase the letters/numbers, make lines thicker and points enlarged. In other words, please make all parts of the figures better visible. Do not use light colors like yellow, for example.

2. Do the authors really mean an oscilloscope in Figure 3? Oscilloscopes were used from 1930s till 1980s. They are old instruments. Normally, they are not used anymore being substituted with modern computerized electronics. Potentially, they might be used now, but it looks strange.

3. Do not use the reference to the German version of a paper published in Angew. Chem. Please change it to the English version, which will be more convenient for international readers. If needed, they can find the German version.

4. The technical details on the electrochemical experiments should be provided. They include:

a) the kind of the electrochemical instrument (name, company, etc.)

b) the reference electrode (e.g., Ag/AgCl or calomel, etc.)

c) the initial and final potentials used in the chronoamperometric measurements

d) the composition of the electrolyte solution, its pH, etc.

e) the kind of the working and counter electrodes (e.g., gold or glassy carbon, etc.)

The suggestions above do not compromise the excellent scientific quality of the paper.

Reviewer #2: The manuscript by Adamatzky et al. presents a new version of electrochemical logic gates. Previously reported systems by Amatore and Warkocz were based on simple redox-active molecular, whereas current submission used complex biopolymers.

Manuscript is technically sound, only minor corrections are necessary.

In the introduction, it would be very beneficial to discuss briefly previous accounts of electrochemical logic gases, e.g. based on those papers:

1. AMATORE, C.; BROWN, A. R.; THOUIN, L.; WARKOCZ, J.-S., Mimicking neuronal synaptic behaviour: Processing of information with 'AND' or 'OR' Boolean logic via paired-band microelectrode assemblies. C. R. Acad. Sci. Paris 1998, 509-515.

2. Amatore, C.; Thouin, L.; Warkocz, J.-S., Artificial Neurons with Logical Properties Based on Paired-Band Microelectrode Assemblies. Chem. Eur. J. 1999, 5, 456-465.

Some other references to electrochemica logic systems may be also added for completness of the picture.

Experimental part misses some precision and crispness.

First of all, thermal processing of aminoacids is a polycondensation process, as small molecules (water, ammonia) are side products (page 2).

For the experimental part, detailed recipes, including amounts of reagents and detailed reaction conditions should be given.

For electrochemical measurement the configuration of the potentiostat should be given (2- or 3-electrode setup, concentration of protenoids, supporting electrolyte...). More detailed information of the experimental setup should be given: types of electrodes, materials of electrodes, electrode spacing, etc. Detailed drawing of photo of the setup would be very beneficial.

Applied voltage values should be given for all experiments.

Authors present detailed analysis of signal statistical features, but use single current measurements for logic analysis - how the time-dependent variability of the signal (Figure 4 and 9) translate into logic performance. It may not be relevant, but should be briefly discussed, as the relevance of signal variability may potentially influence the performance ,especially when different protenoid will have significantly different time scale of variability.

6. PLOS authors have the option to publish the peer review history of their article (what does this mean?). If published, this will include your full peer review and any attached files.

Reviewer #1: No

Reviewer #2: No

---

## [Author Response · Author response to Decision Letter 0]

30 Aug 2023

Response to Reviewer Comments on "Logical Gates in Ensembles of Protenoid Microspheres"

Reviewer 1

All figures must be improved for better visibility (increase the letters/numbers, make lines thicker and points enlarged. In other words, please make all parts of the figures better visible. Do not use light colors like yellow, for example.

We have improved the figures 1, 2, 3, 7, and 11 as per your suggestion. We have increased the font size, contrast, and resolution of the figures to make them more visible and clear. 

Do the authors really mean an oscilloscope in Figure 3? Oscilloscopes were used from 1930s till 1980s. They are old instruments. Normally, they are not used anymore being substituted with modern computerized electronics. Potentially, they might be used now, but it looks strange.

We have updated the figure and corresponding text to reflect the use of a potentiostat for the chronoamperometry measurements.

Do not use the reference to the German version of a paper published in Angew. Chem. Please change it to the English version, which will be more convenient for international readers. If needed, they can find the German version.

 We have changed the citation to the English version of the paper “Molecules That Make Decisions” by Alberto Credi, which was published in Angewandte Chemie International Edition in 2007

The technical details on the electrochemical experiments should be provided. They include:

a) the kind of the electrochemical instrument (name, company, etc.)

The Zimmer Peacock potentiostat Anapot EIS ZP1000080 was utilised for conducting chronoamperometry measurements.

b) the reference electrode (e.g., Ag/AgCl or calomel, etc.)

A two-electrode setup was used for the electrochemical measurements, with iridium-coated stainless steel sub-dermal needle electrodes (Spes Medica S.r.l., Italy) serving as the working and counter/reference electrodes.

c) the initial and final potentials used in the chronoamperometric measurements

The potentiostat applied a constant dc potential (Edc) and measured the current response at 0.1 s intervals for 25,000 s. For the results shown, Edc was 0.01 V, with no initial equilibration period.

d) the composition of the electrolyte solution, its pH, etc.

The electrodes were positioned approximately 10 mm apart in a protenoid solution containing 10 mg/100 ml protenoid in water as the supporting electrolyte.

e) the kind of the working and counter electrodes (e.g., gold or glassy carbon, etc.)

See also b)

Reviewer 2

In the introduction, it would be very beneficial to discuss briefly previous accounts of electrochemical logic gases, e.g. based on those papers:

1. AMATORE, C.; BROWN, A. R.; THOUIN, L.; WARKOCZ, J.-S., Mimicking neuronal synaptic behaviour: Processing of information with 'AND' or 'OR' Boolean logic via paired-band microelectrode assemblies. C. R. Acad. Sci. Paris 1998, 509-515.

2. Amatore, C.; Thouin, L.; Warkocz, J.-S., Artificial Neurons with Logical Properties Based on Paired-Band Microelectrode Assemblies. Chem. Eur. J. 1999, 5, 456-465.

We have added a paragraph to the paper based on the following papers:

1. AMATORE, C.; BROWN, A. R.; THOUIN, L.; WARKOCZ, J.-S., Mimicking neuronal synaptic behaviour: Processing of information with 'AND' or 'OR' Boolean logic via paired-band microelectrode assemblies. C. R. Acad. Sci. Paris 1998, 509-515.

2. Amatore, C.; Thouin, L.; Warkocz, J.-S., Artificial Neurons with Logical Properties Based on Paired-Band Microelectrode Assemblies. Chem. Eur. J. 1999, 5, 456-465.

‘’The field of electrochemical logic gates has seen significant research activity. Amatore and colleagues were among the early pioneers in this area, showcasing the capabilities of paired-band microelectrode assemblies. These assemblies successfully imitate the behaviour of neuronal synapses and are capable of performing Boolean logic operations. Amatore et al. demonstrated that artificial neurons utilising coupled double-band electrodes have the capability to operate as AND and OR logic gates [1],[2]. This is achieved by leveraging the distinctive diffusional cross-talk effects in close proximity to the electrodes. A detailed investigation was conducted on the time responses and theoretical features of these electrochemical logic gates. Expanding upon the encouraging progress made with paired microband electrodes as electrochemical logic gates, our current research focuses on the development of protenoid microsphere-based logic gates. Protenoid microspheres serve as a biomolecular platform that integrates the propagation, transmission, and detection of signals, similar to how natural neurons function.’’

Some other references to electrochemica logic systems may be also added for completness of the picture.

We have added literature from the following papers to our paper:

Zhang, L., Wang, H.X., Li, S. and Liu, M., 2020. Supramolecular chiroptical switches. Chemical Society Reviews, 49(24), pp.9095-9120.

Willner, I., Willner, B. and Katz, E., 2007. Biomolecule–nanoparticle hybrid systems for bioelectronic applications. Bioelectrochemistry, 70(1), pp.2-11.

Valov, I., Waser, R., Jameson, J.R. and Kozicki, M.N., 2011. Electrochemical metallization memories—fundamentals, applications, prospects. Nanotechnology, 22(25), p.254003.

Experimental part misses some precision and crispness.

First of all, thermal processing of aminoacids is a polycondensation process, as small molecules (water, ammonia) are side products (page 2).

We have added a paragraph to page 2 of our paper to clarify this point. The paragraph we added is as follows:

‘’Thermal proteins (proteinoids)~\\cite{fox1992thermal} are synthesized by thermal polycondensation of amino acids. This involves heating a mixture of amino acids to 160-200~\\textsuperscript{o}C under an inert atmosphere, triggering a polycondensation reaction between the amino acids. Rather than a typical polymerization which links monomer units together directly, this is a step-growth polymerization which also generates small molecule byproducts like water and ammonia. The high temperatures cause bifunctional amino acids like glutamic acid to cyclize, which facilitates their role as solvents and initiators for the polycondensation reaction. The end result is a complex mixture of polypeptides with a broad distribution of chain lengths~\\cite{harada1958thermal,fox1992thermal}.’’

For the experimental part, detailed recipes, including amounts of reagents and detailed reaction conditions should be given.

We have revised the sentences to specify the amount of amino acids used:

"The amino acids weighing 5 g in total were heated to their boiling points and mixed together in equimolar amounts."

‘’The resulting mixture was then dissolved in water at a temperature of 80 degrees Celsius, while continuously mixing, to achieve a concentration of 10 mg/100 ml for each protenoid.’’

For electrochemical measurement the configuration of the potentiostat should be given (2- or 3-electrode setup, concentration of protenoids, supporting electrolyte...). More detailed information of the experimental setup should be given: types of electrodes, materials of electrodes, electrode spacing, etc. Detailed drawing of photo of the setup would be very beneficial.

Applied voltage values should be given for all experiments.

Here is a revised paragraph that has been expanded and improved for clarity and coherence:

‘’A two-electrode setup was used for the electrochemical measurements, with platinum and iridium-coated stainless steel sub-dermal needle electrodes (Spes Medica S.r.l., Italy) serving as the working and counter/reference electrodes. The electrodes were positioned approximately 10 mm apart in a protenoid solution containing 10 mg/100 ml protenoid in water as the supporting electrolyte. The potentiostat applied a constant dc potential (Edc) and measured the current response at 0.1 s intervals for 25,000 s. For the results shown, Edc was 0.01 V, with no initial equilibration period. The proteinoids exhibited varying spiking frequency and amplitude when exposed to different applied potentials, specifically positive and negative Edc values. Communication with the protenoids was established using this two-electrode electrochemical cell connected to the potentiostat.’’

Authors present detailed analysis of signal statistical features, but use single current measurements for logic analysis - how the time-dependent variability of the signal (Figure 4 and 9) translate into logic performance. It may not be relevant, but should be briefly discussed, as the relevance of signal variability may potentially influence the performance ,especially when different protenoid will have significantly different time scale of variability.

We have added the following text in the discussion section:

‘’Although our analysis primarily focuses on using single current measurements for implementing logic gates, we acknowledge that the protenoid signals display time-dependent variability, as demonstrated in Figure~\\ref{savdsbfgsndghmjmk,k}. Distinct time scales are observed for the spiking behaviour and noise characteristics of different proteonid types. The variability in the signals over time has the potential to affect the reliability and precision of logic operations, especially when using proteinoids with significantly different dynamics. Logic gating that relies on transient spikes may be impacted if the timing of the spikes is inconsistent. In order to obtain consistent logic inputs from protenoid currents that have a high degree of variability, it may be necessary to employ suitable signal processing or filtering techniques. Further investigation is needed to understand the impact of signal noise and fluctuations on logic accuracy. The present study showcases the use of simple DC signal levels to perform proof-of-concept logic operations. This serves as a starting point for further investigation into more intricate signal processing methods that can help reduce time-dependent variations.’’

---

## [Decision Letter · Decision Letter 1]

1 Sep 2023

Logical gates in ensembles of proteinoid microspheres

PONE-D-23-22314R1

Dear Dr. Mougkogiannis,

We’re pleased to inform you that your manuscript has been judged scientifically suitable for publication and will be formally accepted for publication once it meets all outstanding technical requirements.

Kind regards,

Jianhui Liu

Academic Editor

PLOS ONE

Additional Editor Comments (optional):

Reviewers' comments:

Reviewer's Responses to Questions

**Comments to the Author**

1. If the authors have adequately addressed your comments raised in a previous round of review and you feel that this manuscript is now acceptable for publication, you may indicate that here to bypass the “Comments to the Author” section, enter your conflict of interest statement in the “Confidential to Editor” section, and submit your "Accept" recommendation.

Reviewer #1: All comments have been addressed

Reviewer #2: All comments have been addressed

2. Is the manuscript technically sound, and do the data support the conclusions?

Reviewer #1: Yes

Reviewer #2: Yes

3. Has the statistical analysis been performed appropriately and rigorously? 

Reviewer #1: N/A

Reviewer #2: Yes

4. Have the authors made all data underlying the findings in their manuscript fully available?

Reviewer #1: Yes

Reviewer #2: Yes

5. Is the manuscript presented in an intelligible fashion and written in standard English?

Reviewer #1: Yes

Reviewer #2: Yes

6. Review Comments to the Author

Reviewer #1: The paper was well revised and can be published in the present version. All requests made by the reviewers were satisfied. The paper is indeed good and its urgent publication will be beneficial for the experts in this research area.

Reviewer #2: (No Response)

7. PLOS authors have the option to publish the peer review history of their article (what does this mean?). If published, this will include your full peer review and any attached files.

Reviewer #1: No

Reviewer #2: No

---

## [Editor Report · Acceptance letter]

7 Sep 2023

PONE-D-23-22314R1 

Logical gates in ensembles of proteinoid microspheres 

Dear Dr. Mougkogiannis:

I'm pleased to inform you that your manuscript has been deemed suitable for publication in PLOS ONE. Congratulations! Your manuscript is now with our production department. 

Kind regards, 

on behalf of

Dr. Jianhui Liu 

Academic Editor

PLOS ONE